# Oral Microbiota Linking Associations of Dietary Factors with Recurrent Oral Ulcer

**DOI:** 10.3390/nu16101519

**Published:** 2024-05-17

**Authors:** Yetong Wang, Haiyan Yue, Yuzhou Jiang, Qiumin Huang, Jie Shen, Gulisiya Hailili, Zhonghan Sun, Xiaofeng Zhou, Yanni Pu, Huiling Song, Changzheng Yuan, Yan Zheng

**Affiliations:** 1State Key Laboratory of Genetic Engineering, School of Life Sciences and Human Phenome Institute, Fudan University, Shanghai 200438, China; 2School of Public Health, The Second Affiliated Hospital, Zhejiang University School of Medicine, Hangzhou 310058, China; 3National Clinical Research Center for Aging and Medicine, Huashan Hospital, Fudan University, Shanghai 200040, China; 4Ministry of Education Key Laboratory of Public Health Safety, School of Public Health, Institute of Nutrition, Fudan University, Shanghai 200032, China; 5Department of Cardiology, Shanghai Institute of Cardiovascular Disease, Zhongshan Hospital, Fudan University, 1609 Xietu Road, Shanghai 200032, China

**Keywords:** oral microbiota, plant-based diet, food intakes, recurrent oral ulcer, oral health

## Abstract

Recurrent oral ulcer (ROU) is a prevalent and painful oral disorder with implications beyond physical symptoms, impacting quality of life and necessitating comprehensive management. Understanding the interplays between dietary factors, oral microbiota, and ROU is crucial for developing targeted interventions to improve oral and systemic health. Dietary behaviors and plant-based diet indices including the healthful plant-based diet index (hPDI) were measured based on a validated food frequency questionnaire. Saliva microbial features were profiled using 16S rRNA gene amplicon sequencing. In this cross-sectional study of 579 community-based participants (aged 22–74 years, 66.5% females), 337 participants had ROU. Participants in the highest tertile of hPDI exhibited a 43% lower prevalence of ROU (odds ratio [OR] = 0.57, 95%CI: 0.34–0.94), compared to the lowest tertile, independent of demographics, lifestyle, and major chronic diseases. Participants with ROU tended to have lower oral bacterial richness (Observed ASVs, *p* < 0.05) and distinct bacterial structure compared to those without ROU (PERMANOVA, *p* = 0.02). The relative abundances of 16 bacterial genera were associated with ROU (*p*-FDR < 0.20). Of these, *Olsenella*, *TM7x*, and unclassified Muribaculaceae were identified as potential mediators in the association between hPDI and ROU (all *p*-mediations < 0.05). This study provides evidence of the intricate interplays among dietary factors, oral microbiota, and ROU, offering insights that may inform preventive and therapeutic strategies targeting diets and oral microbiomes.

## 1. Introduction

Recurrent oral ulcer (ROU) is a distressing condition characterized as the most prevalent noninfectious and nontraumatic ulcerative disorder of the oral mucosa [1]. It manifests as recurrent and painful ulcers on the inner surfaces of the oral cavity, and is associated with increased risk of systemic infection, and may impact the quality of life [2]. Diets play important roles in the occurrence and development of ROU.

Emerging evidence has suggested that ROU is associated with dysbiosis of the oral microbiota. Different oral bacterial diversity and structure composition have been observed in patients with ROU compared to that in healthy participants [3,4]. Moreover, alternations in core microbial members were common in the saliva microbiota of patients with ROU [5], such as the decreased abundance of *Streptococcus salivarius* and the increased *Capnocytophaga sputigena*. By metabolizing dietary components, oral microbiota may have direct or indirect effects on the occurrence of ROU [6,7,8].

The oral microbiome has been linked to multiple factors like host age, sex, smoking, alcohol intake, medical treatment, and, especially, the diet habit, which also play important roles in the occurrence and development of ROU [9,10,11,12,13,14,15,16]. For example, a higher frequent consumption of sweet beverages, carbonated beverages, and fried or spicy foods was associated with a higher prevalence of ROU, while higher nut consumption linked to lower ROU prevalence [17,18]. General dietary patterns, sugar and carbohydrate intake, and alcohol consumption may also disrupt the oral microbiome–host balance, potentially precipitating the onset of various oral diseases [19].

However, the potential interplays between dietary factors and oral microbiota, and the impact of such interplays on ROU, are unclear. In the present study, we conducted a cross-sectional study to investigate the associations between dietary factors, oral microbiota, and ROU in adults. Our findings aim to offer practical support for the development of prevention and treatment strategies for ROU.

## 2. Materials and Methods

### 2.1. Study Population and Design

This study was based on the Central China Cohort, a large community-based cohort involving healthy individuals initiated in 2017 in Xinmi City, Henan Province, China [20]. This cohort was established with the aim to investigate the complex interactions among genes, environmental factors, and common diseases, thereby laying the groundwork for the development of strategies for disease prevention and treatment. At the first follow-up visit in 2021, participants were interviewed to gather updated information on sociodemographic characteristics, lifestyle, dietary habits, oral conditions, medical history, and physical measurements. Saliva samples were also collected for microbial assessments at the same time.

In the current cross-sectional study, among the participants who attended the follow-up examination, we excluded those with missing information on the questionnaire (*N* = 8), who did not provide saliva samples (*N* = 10), with <10,000 16S rRNA gene sequencing reads in their saliva samples (*N* = 8), or with implausible dietary energy intake (<500 or >4000 kcal/day, *N* = 7). Eventually, 579 participants were included in the main analysis. The overview of the study design is shown in Appendix A.

### 2.2. Dietary Assessment

Dietary data were collected using a validated 79-item food frequency questionnaire [21], which assessed the consumption frequencies and the portion sizes of food items consumed by the participants over the past 12 months. The average daily intake of each food item was calculated by multiplying its daily frequency and the corresponding portion size. Total energy intake was further calculated based on the China Food Composition Table [22].

Our study primally focused on three plant-based diet indices (PDIs), including the plant-based diet index (PDI), the healthful plant-based diet index (hPDI), and the unhealthful plant-based diet index (uPDI). Specifically, a total of 18 food groups were used to calculate these three indices, which were assigned positive or reverse scores after segregation into quintiles [23], and the overall score for each PDI ranges from 18 to 90, as presented in Appendix A. In analysis of individual food groups, we further focused on 13 main food groups (i.e., refined grains, whole grains, legumes, vegetables, mushroom, fruits, nuts, red meat, processed meat, poultry, fish, dairy, and eggs, Appendix A) that are common to Chinese dietary culture [22,24,25]. Three PDIs and the daily intakes of 13 food groups were divided into tertiles for the following statistical analysis (Appendix A).

### 2.3. Outcome and Covariate Assessment

The information on the presence of ROU was collected by asking the question “Have you experienced oral ulcers?”, with responses recorded as (1) Never, (2) Occasionally, (3) Frequently, and (4) Every day. Participants who reported “Never” were categorized as without ROU, while others were classified as with ROU. Data on detailed demographic, lifestyle, and medical information were obtained from face-to-face interviews. Physical activity from self-reported recalls of occupational, transportation, domestic, and leisure activities during a usual week was abstracted from the long-form international physical activity questionnaire (IPAQ), which has been validated in a Chinese population [26], and we then calculated the daily metabolic equivalent hours of physical activity [27]. Anthropometric parameters, including weight and height, were measured by trained nurses. The body mass index (BMI) was calculated as weight (in kilograms) divided by the square of the body height (in meters). Current smokers were identified as those who had one or more cigarettes every 3 days for at least 6 consecutive months. Common chronic systemic diseases were defined as having at least one of the following prevalent diseases (i.e., those with a prevalence >3% in our study population): hypertension, diabetes, or cardiovascular disease (Appendix A). Oral conditions were defined as having at least one of the following oral conditions: bleeding gums, dental caries, and bad breath.

### 2.4. Saliva Microbiota Profiling

Details on saliva sample collection, microbial DNA extraction, and paired-end 16S rRNA gene sequencing were described previously [20,28]. Briefly, saliva samples were self-collected by participants who received detailed illustrated instructions beforehand, using a saliva collector (CY-98000A, HCY Technology, Shenzhen, China) after gently massaging the cheek. Participants were instructed not to eat, drink, smoke, or chew gum within 30 min before sampling. Microbial genomic DNA was extracted using the DNeasy 96 PowerSoil Pro QIAcube HT Kit (Qiagen, Hilden, Germany) and purified with the QIAcube HT system (Qiagen, Hilden, Germany) according to the manufacturer’s recommendations. The V4 region of 16S rRNA gene was amplified and sequenced on the Illumina Novaseq 6000 PE250 sequencing platform. Sequences of 16S rRNA gene were processed using the Quantitative Insights Into Microbial Ecology 2 (QIIME2, version 2022.2.0) program [29], after splitting raw fastq files according to the barcode sequence, and converted into QIIME2-compatible files. The Divisive Amplicon Denoising Algorithm 2 (DADA2) pipeline was used to perform the quality filtering, denoising, and calling of amplicon sequence variants (ASVs) [30]. Taxonomic assignments were made using the SILVA 138 SSU Ref NR 99 data set [31] via plugins of feature-classifier [32] and classify-sklearn [33]. Four alpha diversity indices were calculated based on the ASV level: abundance-based coverage estimator (ACE), Observed ASVs, Shannon, and Simpson indices. Furthermore, we performed Phylogenetic Investigation of Communities by Reconstruction of Unobserved States 2 (PICRUSt2) to predict the functional potential of microbial communities, which allowed us to gain insights into the functional capabilities of microbial communities without directly measuring their gene expression or functional assays [34].

### 2.5. Statistical Analysis

Differences in characteristics of participants with or without ROU were examined using the Wilcoxon rank sum test and Chi-squared test for continuous variables and categorical variables, respectively. Logistic regression models were performed to examine the associations between dietary factors and ROU, with adjustment for age, sex, current smoking status, physical activity, total energy intake, BMI, and common systematic chronic diseases. The linear trend was examined by adding the median value of each subgroup as a continuous variable in models. Stratified analyses by sex were also performed. Furthermore, we conducted a sensitivity analysis to explore the effects of brushing and oral conditions on the associations of dietary factors with ROU.

Differences in the alpha diversity indices between the two groups were examined using the Wilcoxon rank sum test. A principal coordinate analysis (PCoA) based on Bray–Curtis dissimilarity metrics and robust Aitchison distance metrics, along with a permutational multivariate analysis of variance (PERMANOVA) (999 permutations), was performed to assess the association of overall oral microbial structure with ROU. For taxonomic and functional features, we filtered genus-level microbes and functional pathways with a mean relative abundance of <0.01% and a prevalence of <10%, followed by a centered log-ratio transformation on the relative abundance data, and used a multivariate analysis by linear models (MaAsLin) [35] to examine the associations of bacterial genera and pathways with ROU, adjusted for age, sex, current smoking status, and BMI. All multiple-hypothesis testing was controlled for the false discovery rate (FDR) using the Benjamini–Hochberg method with a significance threshold of *p*-FDR < 0.20. To verify the robustness of the results, we used ANCOM-BC to reanalyze the identified genera. As oral microbiota differs in sex [36], a followed sex-stratified analysis was conducted for the associations of identified genera with ROU. In the sensitivity analysis, we further adjusted for brushing and oral conditions in order to examine their effects on the associations of oral microbiota with ROU.

We performed the mediation analysis to examine whether the associations between dietary factors and ROU were mediated by identified oral microbial features with the *mediation* R package (version 4.5.0) [37]. We assessed the collective mediation effect of oral microbes on the association between hPDI and ROU by creating a bacterial score. Five ulcer-associated genera, namely *Atopobium*, *TM7x*, unclassified Muribaculaceae, *Olsenella*, and *Abiotrophia*, were selected for their correlations with dietary factors, aligning consistently with the correlations observed between bacteria, dietary factors, and ROU. Specifically, *Atopobium* and *TM7x* were positively correlated with ROU and inversely correlated with hPDI, indicating a consistent direction of correlation. Similarly, unclassified Muribaculaceae, *Olsenella*, and *Abiotrophia* had inverse correlations with ROU and positive correlations with hPDI. For the first two genera, a score of 1 was assigned to participants with lower relative abundance than the median of that genus among all participants, while a score of 0 was assigned if it was higher. A reverse value was applied for the last three genera. The final score for each participant ranged from 0 to 5.

All statistical analyses were performed in R version 4.2.1. A *p*-value < 0.05 was considered statistically significant unless otherwise specified.

## 3. Results

### 3.1. Characteristics of Study Participants

Demographic characteristics of the participants are shown in Table 1. Among 579 participants included in the present study, 337 (58.2%) individuals reported having experienced ROU. Compared to the participants without ROU, those with ROU tended to be younger, non-smokers, and have lower hPDI (all *p* < 0.05). There were no significant differences in sex, BMI, total energy intake, PDI, uPDI, physical activity level, common chronic systemic diseases, brushing habit, and oral conditions between the two groups (all *p* > 0.05). The daily intakes of whole grains, red meat, processed meat, poultry, fish, and dairy were different between the two groups (all *p* < 0.05, Appendix A).

### 3.2. Association of Dietary Factors with Recurrent Oral Ulcer

Compared to participants in the lowest tertile, those in the highest hPDI tertile had 43% lower odds of the prevalence of ROU (OR = 0.57, 95%CI: 0.34–0.94, *p =* 0.029), with the adjustment of age, sex, current smoking status, physical activity, total energy intake, BMI, and common chronic systemic diseases (Figure 1). No significant association of the PDI and uPDI with ROU was observed (*p* > 0.05, Figure 1). The observed associations were generally robust with further adjustment for the brushing habit and oral conditions (Appendix A). In the secondary analysis of individual food groups, participants in the highest tertile of red meat and egg intakes were associated with 88% and 73% higher odds of ROU, respectively (Figure 1); the ORs of ROU comparing the highest versus lowest tertile were 1.88 (95%CI: 1.12–3.11, *p =* 0.015) for red meat and 1.73 (95%CI: 1.06–2.83, *p* = 0.028) for eggs. Furthermore, significant sex differences were observed for the association of refined grains with ROU, which was only significant among females (*p*-interaction = 0.004, Appendix A).

### 3.3. Association of Oral Microbiota with Recurrent Oral Ulcer

For the alpha diversity analysis, participants with ROU had lower Observed ASVs of microbial communities than their counterparts (*p* = 0.03), though no statistically significant differences in other diversity indices were observed between the two groups (Figure 2A). Furthermore, the oral microbial community structure was significantly different between the two groups based on the Bray–Curtis dissimilarity metrics and robust Aitchison distance metrics (both *p* < 0.05, PERMANOVA, Figure 2B). The sex-stratified analysis revealed that only males showed a significant link between Observed ASVs and ROU (*p* < 0.05, Appendix A), and differences in the oral microbial community structure between the two groups were observed for both sexes (the robust Aitchison distance for males and both Bray–Curtis and robust Aitchison distances for females; *p* < 0.05, PERMANOVA, Appendix A).

Among the 89 detected genera, 16 genera were associated with the prevalence of ROU with adjustment for age, sex, current smoking status, and BMI (*p*-FDR < 0.20, Figure 2C). Specifically, the relative abundances of five genera (e.g., *Leptotrichia*, *Actinomyces*, and *Atopobium*), which mainly belong to the phyla Fusobacteria and Actinobacteria, were positively associated with ROU. Conversely, the relative abundances of 11 genera (e.g., *Wolinella*, *Olsenella*, and *Abiotrophia*) were inversely associated with ROU. Eight genera remained significant after adjusting for brushing and oral conditions as additional covariates (*p*-FDR < 0.20, Appendix A). To verify the robustness of the results, ANCOM-BC was used to validate the identified genera, and eight genera retained a consistent association with ROU (*p* < 0.05, Appendix A). In addition, ROU-associated genera were not exactly the same in different sexes (*p* < 0.05, Appendix A). No significant interaction between sex and oral microbiota was observed for the associations mentioned above (all *p*-interactions > 0.05, Appendix A).

Among the 262 included predictive functional pathways, 45 were found to be associated with ROU status after adjusting for the potential covariates (*p*-FDR < 0.20, Appendix A and Appendix A). Among them, 26 pathways were more enriched among participants with ROU (Appendix A), whereas 19 pathways were inversely associated with ROU (Appendix A). Generally, these pathways were mainly involved in nucleoside and nucleotide degradation/biosynthesis, cofactor/carrier/vitamin biosynthesis, and amino acid biosynthesis.

### 3.4. Associations between Dietary Factors and Identified Microbial Features

We proceeded to investigate the relationships between dietary factors and ROU-associated bacterial genera. The relative abundances of nine genera differed between the extreme tertiles of hPDI. *Atopobium* and *TM7x*, which were positively associated with ROU, were lower in the highest hPDI tertile comparing with the lowest tertile (all *p* < 0.05, Figure 3A). The relative abundances of three genera (*Wolinella*, *Olsenella*, and *Streptobacillu*) and one genus (Rikenellaceae RC9 gut group) were significantly different between the extreme tertiles of red meat and egg intakes, respectively (all *p* < 0.05, Figure 3B,C). In addition, the genus *Olsenella* was observed to be more abundant in participants with a higher hPDI score and less abundant in participants with higher red meat intake (both *p* < 0.05, Appendix A). No such trends in genus abundance were observed along with egg intakes (Appendix A). Consistent with the previous results, associations between dietary factors and identified genera varied by sex (Appendix A). For instance, the association between Rikenellaceae RC9 gut group and hPDI was only significant among females, while the link between *Bergeyella* and red meat was solely observed among males. No such trends in genus abundance were observed along with egg intakes (Appendix A). In terms of functional features, 11 pathways, such as PANTO-PWY (phosphopantothenate biosynthesis) and P164-PWY (purine nucleobases degradation), significantly differed between the extreme tertiles of hPDI (Appendix A).

### 3.5. Oral Microbiota Might Mediate the Association of Dietary Factors with Recurrent Oral Ulcer

In the mediation analysis, we observed that the association between hPDI and ROU was partly mediated by 3 genera (*TM7x,* unclassified Muribaculaceae, and *Olsenella*) of the 16 ulcer-associated genera, with a mediation proportion of 10.77%, 17.23%, and 11.04%, respectively (all *p*-mediations < 0.05, Figure 4A–C). Genus *Olsenella* potentially mediated about 14% of the association between red meat intake and ROU (*p*-mediation = 0.02, Figure 4D). Of note, we observed that the composite bacterial score based on the five ulcer-associated genera potentially mediated 25.50% of the association of hPDI with ROU (*p*-mediation < 0.001, Figure 4E). Among the 45 ROU-associated functional features, only one pathway (PANTO-PWY: the biosynthesis of phosphopantothenate) partly mediated the association of hPDI (mediation proportion: 5.62%) and red meat (mediation proportion: 14.50%) with ROU (both *p*-mediations < 0.05, Appendix A).

## 4. Discussion

In the presented study, we comprehensively examined the associations between dietary factors, oral microbiota, and ROU status. We observed that participants with a higher hPDI score were less likely to experience ROU, whereas those with higher intakes of red meat and eggs tended to experience ROU. We also revealed significant differences in the oral microbial features between participants with or without ROU, with 16 genera significantly associated with ROU. In addition, several microbial features associated with ROU were found to potentially mediate the associations of hPDI and red meat intake with ROU, suggesting that the oral microbiota may be involved in the potential role of diets in ROU.

In our study, participants with a higher hPDI score were less likely to experience ROU, which is in line with previous studies reporting the beneficial role of plant foods in periodontal health [38,39,40,41]. The plant-dominant food components contribute to reduced levels of oxidative stress and inflammation, as well as elevated concentrations of antioxidant and anti-inflammatory biomarkers, known for their beneficial effects on oral health [42]. Until now, only a few studies investigated the associations between specific food groups and the development of ROU. For example, previous cross-sectional studies on ROU populations have shown that higher nut consumption could reduce the risk of ROU [17]. Some studies suggested that a higher intake of healthy plant foods containing trace elements, including grains and fruits, may be beneficial for the prevention or treatment of ROU. In addition, limited studies have explored the associations between specific animal foods (e.g., red meat and eggs) and oral diseases. Despite this, there have been reports suggesting that an excessive intake of meat may increase the risk of oral diseases [43], and an excessive consumption of red meat can promote oxidative stress and inflammatory responses in the body due to its high fat and cholesterol content [44,45]. Saturated fatty acids are typically found in processed meats, dairy products, and eggs, which are positively associated with the development of periodontal disease [46].

A few studies observed no differences in alpha diversity indices between individuals with ROU and healthy controls [5,47], although our finding and some others indicated a trend of differing alpha diversity in ROU populations [48]. Regarding the structural changes in oral microbiota in individuals with ROU, in line with our findings, related studies have consistently observed significant differences in beta diversity between individuals with and without ROU [3,4].

We observed that several genera negatively correlated with ROU primarily belonged to the Firmicutes phylum, which is also consistent with previous research [47]. We further observed that oral bacteria positively associated with ROU included *Leptotrichia*, *Actinomyces*, *Atopobium*, *TM7x*, and *Corynebacterium*. Most of these genera were previously implicated in oral health, though direct evidence linking them to ROU is lacking. *Leptotrichia* was found to be associated with active oral ulcers [3], and previous investigations have isolated *Leptotrichia* from blood cultures of patients with lesions in the oral mucosa [49]. Emerging evidence has shown that TM7 bacteria are positively associated with inflammatory mucosal diseases [50] and periodontal mucosal infections [51]. Therefore, an increase in the relative abundance of TM7 may indicate the occurrence of ROU. *Atopobium* may play an important role in caries production [52,53], while *Actinomyces* has been implicated in causing oral abscesses [54] and was correlated to the presence of gingivitis and a periapical lesion [55]. Studies have also found associations between *Corynebacterium* and periodontal health [52,56]. Regarding the genera inversely correlated with oral ulcers, *Abiotrophia defectiva* was considered as a beneficial bacterial species for periodontal health [57,58], and *Peptococcus* has been identified as a taxon correlated with periodontitis [59]. However, associations of the remaining taxa with oral health were less frequently studied. Furthermore, we observed the associations of both dietary factors and oral microbiota with ROU varied by sex, which is consistent with previous studies, suggesting that sex plays significant roles in the interplays of nutritional patterns and oral microbiota with oral health [60,61].

Our study has several limitations. First, the current analysis relied on questionnaires to collect data on dietary intake and oral conditions. Despite that the validation study of our food frequency questionnaire showed robust reliability and validity [21], recall bias relying on participants’ ability to recall their food intake frequency and quantity over a specific period is quite possible. Meanwhile, we did not collect information on visits to dental professionals or oral hygiene practices or medication. Second, 16S RNA gene sequencing lacks resolution to determine functional and species-level association. Third, as an observational cross-sectional study, we cannot make causal inferences on the association among dietary factors, oral microbiota, and ROU. Despite these limitations, this study stands as the first large-scale population-based research to thoroughly examine the complex relationship between dietary habits, oral microbiota, and ROU, providing valuable insights for enhancing oral health. Understanding these interplays is pivotal for developing targeted interventions or personalized nutrition aimed at improving both oral and systemic health outcomes.

Further efforts should encompass longitudinal studies, intervention trials, and the integration of mechanistic investigations to advance our understandings and ultimately lead to more effective preventive and therapeutic strategies for enhancing oral health. Future research should also investigate natural agents like probiotics, paraprobiotics, postbiotics, and ozonated substances for their ability to modulate the microbiota and reduce lesion occurrence [62,63]. Additionally, an assessment of factors such as oral hygiene, oral health status, and medication use is vital to minimize confounding effects, providing a clearer understanding of the interplay between ROU, dietary factors, and oral microbiota. Employing shotgun metagenomic sequencing and conducting prospective cohort studies or animal research are warranted to unravel these complex associations and mechanisms.

### Clinical Relevance

The rationale for this study lies in its potential to expand our understanding of how dietary factors and the oral microbiota interact to influence ROU. By identifying modifiable risk factors and potential microbial mediators in the association between diet and ROU, this study aims to contribute to the development of more effective strategies for the prevention and treatment of this condition.

This study found that adherence to a healthful plant-based diet is associated with a reduced prevalence of ROU, and individuals with ROU have lower oral bacterial richness and distinct microbial compositions. Furthermore, certain specific bacterial genera may partly mediate the relationship between dietary intakes and ROU.

Our findings suggest that potential dietary adjustments and microbiome-focused interventions could mitigate ROU risk, guiding future preventive and therapeutic strategies.

## 5. Conclusions

Our findings underscore the significance of proactive prevention in clinical practice. In our study, individuals with a higher dietary hPDI score were less likely to experience ROU, and ROU-related bacterial taxa partially mediated the association of dietary intakes with ROU. These findings shed light on the complex interplay between dietary factors, oral microbiota, and ROU. These insights contribute to a paradigm shift in clinical thinking—away from reactive measures and towards proactive, dietary, and microbiome-based strategies that prioritize prevention in the treatment of recurrent oral ulcers.

## Figures and Tables

**Figure 1 nutrients-16-01519-f001:**
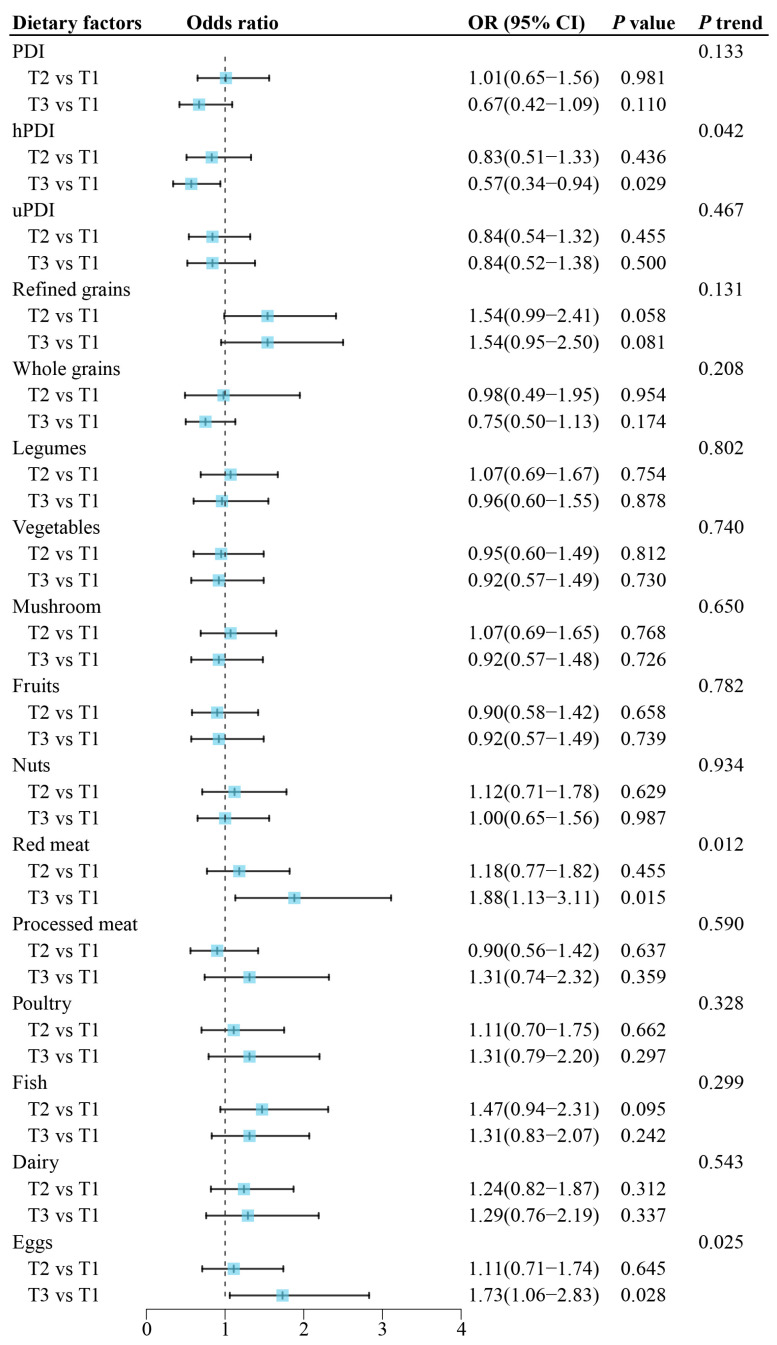
Associations of dietary factors with recurrent oral ulcer. Odds ratios were derived from logistic regression models for tertile 2 (T2) and tertile 3 (T3) of included dietary factors using tertile 1 (T1) as the reference group. Covariates included age, sex, current smoking status, physical activity, total energy intake, BMI, and common chronic systemic diseases. PDI—plant-based diet index, hPDI—healthful plant-based diet index, uPDI—unhealthful plant-based diet index.

**Figure 2 nutrients-16-01519-f002:**
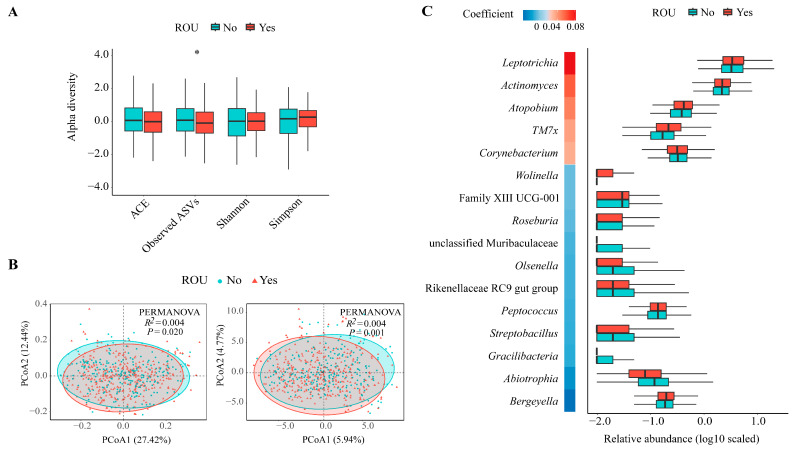
Diversity estimates and the relative abundance of oral microbiota associated with recurrent oral ulcer. (**A**) Comparisons of ACE, Observed ASVs, Shannon, and Simpson indices in the oral microbiota between participants with or without ROU. * *p* < 0.05. All four indices were z-score standardized for visualization. (**B**) PCoA based on the Bray–Curtis distances (left) and robust Aitchison distances (right) of the oral microbial communities between participants with or without ROU. (**C**) The heatmap shows the coefficients of specific genera derived from the MaAsLin analysis with *p*-FDR < 0.20. The box plots show the relative abundances (log-10 transformation) of differentially abundant genera based on ROU status. ROU—recurrent oral ulcer. ASVs—amplicon sequence variants. ACE—abundance-based coverage estimator. PCoA—principal coordinate analysis. MaAsLin—microbiome multivariable associations with linear models.

**Figure 3 nutrients-16-01519-f003:**
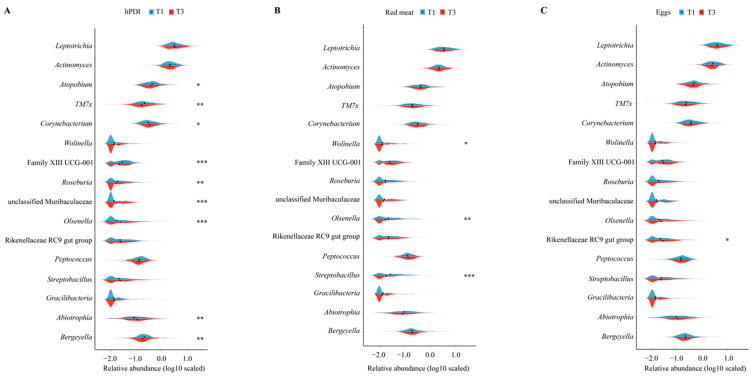
The association of hPDI (**A**) and red meat (**B**) and egg intakes (**C**) with ulcer-associated genera. The relative abundance difference of 16 identified genera between extreme tertiles (T1 and T3) of hPDI, red meat and egg intakes. * *p* < 0.05, ** *p* < 0.01, *** *p* < 0.001. hPDI—healthful plant-based diet index.

**Figure 4 nutrients-16-01519-f004:**
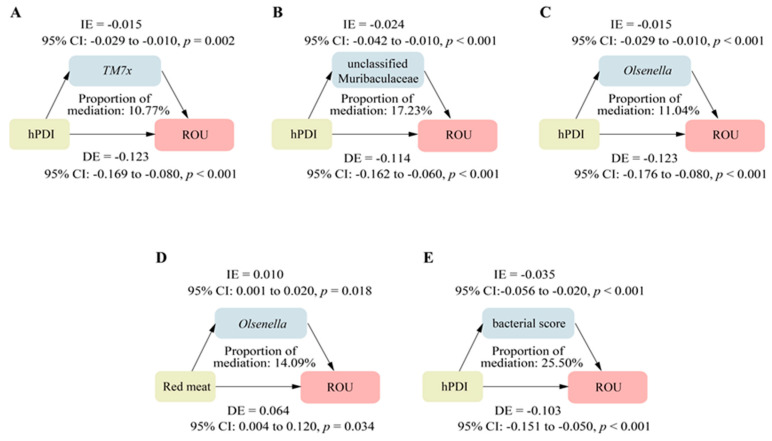
The associations’ mediation effect of oral microbial genera among the association between specific dietary factors and recurrent oral ulcer. (**A**–**C**) The associations’ mediation effect of *TM7x,* unclassified Muribaculaceae, and *Olsenella* on the association between hPDI and ROU, respectively. (**D**) The associations’ mediation effect of *Olsenella* on the association between red meat and ROU. (**E**) The associations’ mediation effect of the composite bacterial on the association between hPDI and ROU. The direct effect (DE) refers to the effect of dietary factors on ROU when bacterial genera were held at a level. The indirect effect (IE) refers to the impact of dietary factors on ROU through bacterial genera. The proportion of mediation represents the percentage of the associations between dietary factors and ROU that is explained by bacterial genera. ROU—recurrent oral ulcer, hPDI—healthful plant-based diet index, IE—indirect effect, DE—direct effect.

**Table 1 nutrients-16-01519-t001:** Characteristics of study participants based on the presence of recurrent oral ulcer.

Characteristics	Recurrent Oral Ulcer		*p*-Value *
	Yes (*N* = 337)	No (*N* = 242)	
Age (year)	43.0 (32.0, 54.0)	54.0 (48.0, 61.8)	<0.001
Male (*n*, %)	104 (30.9)	90 (37.2)	0.521
BMI (kg/m^2^)	24.6 (22.3, 26.8)	25.0 (23.1, 27.8)	0.211
PDI	48.0 (44.0, 53.0)	49.0 (45.0, 54.0)	0.149
hPDI	56.0 (50.0, 61.0)	60.0 (56.0, 64.0)	0.023
uPDI	56.0 (51.0, 62.0)	57.0 (50.2, 62.0)	0.609
Physical activity level			0.186
Low	102 (30.3)	91 (37.6)	
Medium	103 (30.6)	91 (37.6)	
High	132 (39.2)	60 (24.8)	
Common chronic systemic diseases	124 (36.8)	143 (59.1)	0.084
Current smoking	31 (9.2)	42 (17.4)	0.001
Total energy intake (kcal/day)	1842 (1512, 2164)	1883 (1570, 2175)	0.888
Brushing (≥2 times/day)	163 (48.4)	95 (39.3)	0.964
Oral conditions	322 (95.5)	219 (90.5)	0.414

Data are presented as the median (interquartile ranges) for continuous variables, and n (percentage) for categorical variables. The percentage sum of some cells is not equal to 100 because the percentage was rounded to retain decimal places. * *p*-Values were based on the Wilcoxon rank sum test for age and logistic regression for other variables with adjustment for age. PDI—plant-based diet index, hPDI—healthful plant-based diet index, uPDI—unhealthful plant-based diet index, BMI—body mass index.

## Data Availability

Sequencing data during the current study can be viewed in the NODE database (https://www.biosino.org/node/project/detail/OEP004135, accessed on 5 May 2023) and are available upon acceptance of the publication.

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
