# Peer review of "Oral Microbiota Linking Associations of Dietary Factors with Recurrent Oral Ulcer"

_nutrients, 2024, doi:10.3390/nu16101519_

Round 1
Reviewer 1 Report
Comments and Suggestions for Authors
Review - Nutrients-2989937
This manuscript describes a cross-sectional study that examines the relationship between diet, the oral microbiome, and recurrent oral ulcers (ROU) in a large cohort of adults. The study demonstrates (i) and inverse association between healthy plant based dietary patterns and ROU, (ii) an association between specific salivary microbial features and ROU, and (iii) potential microbial mediators in the association between diet and ROU. A major strength of the study is access to a large collection of saliva samples with covariate data. Although this study provides interesting insights into the interplay between these factors, it does lack some critical details in the methodology and would greatly be strengthened by incorporating multiple microbial testing methods to ensure robust interpretation of results.
Introduction
- As done with the diet section, the microbiome background section (line 46-48) would be strengthened by providing specific examples of the changes in ROU patients reported in the literature (e.g. is diversity decreased or increased with ROU, and what are the core microbial members that decline with ROU?).
Methods
Section 2.1
- To give context to the study, a brief description of the cohort should be provided, including why the cohort was set up, recruitment setting (e.g. dental clinic, hospital, general public), and general health status (e.g. healthy, or specific disease).
- given this is a cross sectional study, the wording “the follow up examination” (line 63-54 ) is confusing. Were the samples and questionnaire data all collected at the same visit? Or were there separate follow-up visits? This should be made clear.
Section 2.2
-add the number of items in the FFQ
-please provide additional information on the scoring including the total score for each PDI as well as for of the 13 components. Also include the amounts or daily intakes that correspond to a min or max score (e.g. daily intake of no vegetables serving scores 0, whereas 4 daily servings scores 5). See Table 1 in https://cdnsciencepub.com/doi/full/10.1139/apnm-2023-0018
Section 2.3
-why were only 3 chronic diseases included?
-covariates should include the oral conditions and medical history as noted in section 2.1. In particular, authors should include some measures of oral health, hygiene, and/or oral conditions. And if available, medication used for ROU or other oral conditions should be included.
Section 2.4
Although saliva samples have been described previously, some minimal information on sample collection should be provided here, including sample collection devices, if the saliva collection was stimulated/unstimulated, and if/how long participants avoided food and drink before providing a sample.
-Please provide the version of SILVA and taxonomic information varies among versions.
Section 2.5
- It is unclear why specific variables were chosen as covariates while others were left out. Since this study focuses on the oral microbiome, it is important to include relevant covariates in statistical models including oral hygiene practices/oral health/visits to dental professionals and oral ROU medications.
-With such a large sample size, a sex-stratified analysis should be included since both dietary habits and the oral microbiome are known to differ by sex https://pubmed.ncbi.nlm.nih.gov/34266509/; https://www.ncbi.nlm.nih.gov/pmc/articles/PMC9843272.
-As done with alpha diversity, additional beta-diversity metrics should be included (e.g. weighted UniFrac or robust Aitchison’s distance).
-Differential abundance tools vary in their consistency and power to detect differences (https://www.ncbi.nlm.nih.gov/pmc/articles/PMC8763921/ ; https://www.ncbi.nlm.nih.gov/pmc/articles/PMC9488820/). An additional differential abundance tool(s) (e.g. ANCOM, ALDEx2) should be included to ensure robustness of findings and interpretation of results.
-line 138 - I think this should be mediation.
-For the mediation analysis, it is unclear why only select features were considered? how were these selected?
Results
Section 3.1
-line 158 - correct the descriptive statement about ROU participants and current smokers. There should be fewer current smokers or a lower proportion of current smokers.
-Please include information on oral health, hygiene, and medications in the text and in Table 1. It is noted in section 2.1 that data on oral conditions was captured yet it is never reported in this manuscript.
-It is noted that the prevalence of common chronic diseases is lower in those with ROU. But “common chronic diseases” in the current study only reflect 3 diseases and none of them are oral. Also, is it common that the ROU patients have less common chronic diseases? Or is it possible that only including 3 conditions means the authors are overlooking a large number of conditions that may be more prevalent in patients with ROU?
Section 3.3
- “These pathways were mainly involved in nucleoside and nucleotide degradation/biosynthesis, cofactor/carrier/vitamin biosynthesis, and amino acid biosynthesis”. This sentence refers to Figure S2 however these pathways are not clear in the figure. Please add pathway descriptions of the PICRUSt2 output /specific names of the predicted pathway abundances to Figures S2 and S6.
Discussion
-line 308 to 311 - How can these findings be used to enhance oral health? What research is needed next? Please add future directions of this research.
Comments on the Quality of English Language
good. minor edits only (e.g. line 138 meditation).
Author Response
We have made all requested changes which are highlighted in red throughout the revised manuscript (track-changes version). Please find our point-by-point responses to the reviewers’ comments below.

Reviewer 2 Report
Comments and Suggestions for Authors
I had the pleasure of reviewing this interesting study.
I believe this study is important in attempting to determine the relationship between recurrent oral ulcers (ROU), dietary factors, and the oral microbiome. The manuscript is well organized and well organized. However, in order to improve the completeness of the paper, some revisions are as follows:
In the introduction section, please add the need for the study.
In the Methods section, please note that the study started in 2017, but please revise to the correct data collection period.
In the conclusion section, the findings are interesting.
In the Discussion section, it would be more interesting for readers if the clinical strengths of this study were highlighted in separate subheadings.
Comments on the Quality of English Language
Author Response

(The authors gave the same response as above.)

Reviewer 3 Report
Comments and Suggestions for Authors
Manuscript of interest to the dental sector, especially oral medicine, requires a major revision.
Abstract: Well described with the results clearly outlined.
Keywords: very long, add specific ones that are registered on MeSH but short.
Introduction: very brief, add all the factors that influence the oral microbiota and the differences in the various age groups but above all based on periodontal and/or implant pathologies (Scribante et al).
Materials and methods well described.
Very confusing results, reorganize the tables to make them accessible to the reader by highlighting the statistically significant data.
Discussion: add as future objectives the evaluation of any natural substances that regulate the microbiota and reduce the incidence of lesions, such as probiotics, paraprobiotics, postbiotics, ozonated substances (Butera et al.)
Conclusion: prevention is the first winning weapon therefore the clinician's mentality must be changed by placing the focus on proactive action.
Bibliography: add required references.
Author Response

(The authors gave the same response as above.)

Round 2
Reviewer 1 Report
Comments and Suggestions for Authors
The authors have provided a greatly improved version of their manuscript which includes critical details to the methods section and additional results to demonstrate the robustness of the findings.
The only remaining concern is about the transparency to the reader about the chronic diseases selected. Readers need to understand that this is a relatively healthy cohort and why only select diseases were included. Please include a brief statement that indicates that only diseases with a prevalence over 3% were included in the analysis and provide the actual prevalence of the selected diseases.
Author Response

(The authors gave the same response as above.)

Reviewer 3 Report
Comments and Suggestions for Authors
https://pubmed.ncbi.nlm.nih.gov/38047757/
I would only add this quote, you only entered the paraprobiotics
Author Response

(The authors gave the same response as above.)
